# Adverse Human Health Effects of Chromium by Exposure Route: A Comprehensive Review Based on Toxicogenomic Approach

**DOI:** 10.3390/ijms24043410

**Published:** 2023-02-08

**Authors:** Dong Yeop Shin, Sang Min Lee, Yujin Jang, Jun Lee, Cheol Min Lee, Eun-Min Cho, Young Rok Seo

**Affiliations:** 1Institute of Environmental Medicine for Green Chemistry, Department of Life Science, Dongguk University Biomedi Campus, 32 Dongguk-ro, Ilsandong-gu, Goyang-si 10326, Republic of Korea; 2Insilicogen Inc., Suwon 441813, Republic of Korea; 3Department of Nano, Chemical & Biological Engineering, College of Natural Science and Engineering, Seokyeong University, Seoul 02173, Republic of Korea

**Keywords:** chromium, route of exposure, gene expression profiling, molecular network analysis, go term, gene–gene interaction network

## Abstract

Heavy metals are defined as metals with relatively high density and atomic weight, and their various applications have raised serious concerns about the environmental impacts and potential human health effects. Chromium is an important heavy metal that is involved in biological metabolism, but Cr exposure can induce a severe impact on occupational workers or public health. In this study, we explore the toxic effects of Cr exposure through three exposure routes: dermal contact, inhalation, and ingestion. We propose the underlying toxicity mechanisms of Cr exposure based on transcriptomic data and various bioinformatic tools. Our study provides a comprehensive understanding of the toxicity mechanisms of different Cr exposure routes by diverse bioinformatics analyses.

## 1. Introduction

Heavy metals are defined as metals with relatively high density, atomic weight, or atomic number. Their widespread industrial, domestic, agricultural, medical, and technological applications have raised serious concerns about the environmental impacts and potential human health effects of these elements [1,2]. Due to the ubiquitous characteristic of heavy metals, human exposure can occur through inhalation, ingestion, and dermal routes. Because of the high degree of toxicity, toxicological studies of various heavy metals have been actively conducted and the metals are managed through regulation. Heavy metal exposure also induced diverse cancer. The International Agency for Research on Cancer (IARC) designated arsenic, cadmium, chromium, and nickel as Group 1 carcinogens [3] (i.e., they are carcinogenic to humans).

Chromium is an important micronutrient that is involved in carbohydrate, lipid, and protein metabolism [4]. Cr is an important metal in pigment manufacturing, painting, metal plating, wood treating, and leather tanning, and it is used in metallurgy and refractory material production, among various other applications [5]. Cr is abundant in the Earth’s crust, in which it occurs at a level of 100 μg/g, and its toxicity depends on its chemical state [6,7]. Its two most prevalent oxidation forms are trivalent and hexavalent, but whereas trivalent Cr is largely non-toxic, hexavalent Cr is toxic and carcinogenic in living organisms. The salt form of trivalent and hexavalent Cr compounds, called chromate, generates toxic dust during manufacturing processes [8]. Many studies have investigated the adverse effects of Cr dust on chromate production workers [9,10]. Furthermore, industrial wastes containing Cr are known as major sources of soil and water pollution and, hence, can have a severe impact on public health.

According to the National Institute for Occupational Safety and Health (NIOSH), adverse effects of Cr exposure vary depending on their exposure paths: dermal contact, inhalation, and ingestion [11]. The general population is exposed to Cr from the ingestion of Cr-contaminated food or drinking water, mainly around a specific area contaminated with Cr for a long time [12]. In contrast to exposure via the ingestion route, Cr inhalation and skin contact exposure occurs occupationally, such as in tanneries or timber industries [13]. In addition to the level of exposure, the toxic effects are highly dependent on the exposure path [14].

Because of increased attention to health and biology, a large collection of toxicological data and literature are available. Data-mining techniques and biological analysis tools are useful for organizing “big data” to understand the biological effects of heavy metals [15]. Molecular network analysis is one way to provide a comprehensive view of heavy metal-induced adverse effects and related diseases or potential marker proteins [8,16]. This approach can be undertaken using molecular network analysis software to construct a network about toxic effects on genes and cellular processes and provide disease information.

Gene expression profiling provides the expression level of thousands of mRNAs at the same time, allowing insight into cellular conditions [17]. By analyzing gene expression profiling data, the variation of mRNA level can be used to detect cellular status under certain conditions, such as medicines, toxic substances, or disease status, and explore the underlying mechanisms of chemical toxicity. In addition, network-based hub gene selection and cellular processes contribute to the prediction of specific protein markers for toxicity.

Protein markers are useful for determining whether exposure to certain harmful substances has occurred or to predict the occurrence of certain diseases, but reliance on a single function of a gene/protein to predict toxic effects may lead to misinformation, particularly if it exhibits its toxic effect only after prolonged exposure. Functional enrichment is a widely used tool to identify trends in large-scale biological datasets [18]. It can integrate multiple functional information from various sources to provide a list of genes and the biological processes significantly enriched in these genes [19]. One of the most popular databases of functional enrichment analysis is Gene Ontology, providing information about biological processes, cellular components, and molecular functions.

In this study, we explore the toxic effects of Cr exposure through three exposure routes: dermal contact, inhalation, and ingestion. We summarize the sources of Cr exposure and a undertake a network analysis of public gene expression big data to analyze the toxic effects (Figure 1). A gene ontology term analysis was conducted to predict the biological functions of differentially expressed genes (DEGs), and the toxic effects of Cr exposure were explored through a comparison of the results (Figure 1b). Finally, we present a common and specific toxic effect from all three routes of Cr exposure.

## 2. Adverse Effects of Cr on Skin

Dermal exposure to Cr for the general public can occur from dermal contact with certain consumer products or soils that contain Cr. Cr compounds can be present in cosmetic products and leather products in the form of both Cr(III) and Cr(VI) [20]. Construction workers can be exposed to Cr (VI) present in cement [21]. Cr can penetrate human skin to some extent, especially if the skin is damaged. Penetrated Cr can bind to immune cells of the skin, causing dermal toxicity [22]. Cr exposure can lead to sensitization and elicitation of contact dermatitis by forming Cr-protein conjugates. In addition, Cr exposure can cause skin irritation that may result in ulceration [22].

To understand the toxic mechanisms of Cr dermal contact, we conducted a molecular network analysis with Pathway Studio software (Version 12.5.0.2) using the gene expression profile dataset obtained from the public gene expression database. We used the microarray dataset GSE16394 derived from the study of human dermal fibroblast exposure to in vitro potassium dichromate from the Gene Expression Omnibus (GEO) database (https://www.ncbi.nlm.nih.gov/geo/ (accessed on 1 November 2022)). Figure 2a shows that exposure to Cr and Cr compounds through dermal contact mainly induces DNA damage, metastasis, and immune system disorders. Skin diseases, such as skin damage and ulceration, allergic reactions, and hypersensitivity, seem to be the effects of skin exposure to Cr. Rudolf et al. revealed that Cr exposure led to DNA damage in human dermal fibroblasts [23]. Our molecular network analysis result showed that dermal exposure to Cr was also related to immune system responses. Thomas et al. identified that chromate exposure caused monocyte and dense T-cell infiltration, pointing to a delayed-type immune reaction [24]. In another study, mRNA expression of IL-4, IL-6, and IFN-Q was increased in skin tissues exposed to chromate [24]. Moreover, dermal exposure to Cr has been implicated in immediate-type or type I hypersensitivity reactions. Cases of asthma with an immediate or delayed response have been observed following occupational exposure to Cr [25].

A molecular pathway analysis has advantages in visualizing major cellular processes and disease information based on hub genes, but biases may exist when determining the toxic effects of Cr from dermal exposure because of confounders in the study and references. We identified the biological processes (BP), cellular components (CC), and molecular functions (MF) associated with dermal exposure to Cr by Gene Ontology (GO) term analysis using all DEGs (Figure 2b). Enriched GO pathways were determined based on the cutoff criterion of adjusted *p*-value < 0.05. By GO term analysis, we explored the functions of genes whose expression changed after dermal exposure to Cr. The results showed enrichment of several BP terms, including RNA modification, transfer RNA (tRNA) processing, non-coding RNA processing, sterol biosynthesis process, and cholesterol biosynthesis process. According to Pavesi and Moreira, non-coding RNA induced by Cr exposure could trigger epigenetic modifications that may lead to DNA damage and cancer [26]. Guo et al. revealed that skin cell exposure to Cr induced elevated cholesterol biosynthesis and cytotoxic effects [27]. DNA replication and DNA unwinding in DNA replication were major BP terms suppressed by dermal exposure to Cr. Intrinsic apoptotic signaling pathways were also suppressed. Although it is known that Cr increases replication stress-associated DNA damage [28], the association with the decrease in DNA replication is not yet known. Among the CC terms, the transcriptional regulator complex and RNA polymerase II regulator complex were enriched. This was consistent with previous results that Cr disrupted the binding of the transcriptional factor CTCF and subsequently affected transcription regulation [29]. The CMG helicase complex was significantly suppressed. CMG plays a crucial role in the initiation of DNA replication by unwinding duplex DNA. Among the MF terms, GTPase regulatory activity (GTPase regulator activity and small GTPase binding) was enriched, whereas single DNA helicase activity was a downregulated pathway. Enriched GTPase regulatory activity inhibits protein synthesis [30]. DNA helicase activity is associated with nucleotide excision repair; suppressed DNA helicase activity blocks the repair of DNA adducts induced by Cr exposure [31]. As a result of GO term analysis, the impaired synthesis of various biomolecules and the inhibition of DNA damage repair are predicted to be the main toxic effects of Cr dermal contact.

A network-based gene–gene interaction analysis was performed to select the hub genes associated with Cr dermal contact. The gene–gene interaction network was constructed using Cytoscape (Version 3.7.2) to analyze the topological parameters among the genes (Figure 2c). The size of each node represents the edge degree, and the color of the node represents the betweenness centrality; a bright color indicates high centrality. Finally, we identified the hub genes that may contribute to the dermal effects of Cr exposure, and the core network was identified through bioinformatics approaches (Figure 2d). CXCL8 is a chemokine secreted when keratinocytes are exposed to allergens or irritants [32]. CXCL8 contributes significantly to disease-associated processes, including tissue injury, fibrosis, angiogenesis, and tumorigenesis in the skin. PTGS2 participates in the inflammatory pathway and plays an important role in cutaneous aging and damage, as well as in the activation of acute and chronic neurodegenerative pathologies associated with neuronal death [33]. Overexpression of FOS can lead to the development of tumor cells [34]. HMOX1 is the most consistently observed genetic marker induced by skin sensitizers; sensitizers activate the Keap1–Nrf2 pathway to promote HMOX1 expression [35]. Hao et al. revealed that ATF3 might promote skin cancer cell proliferation and enhance skin keratinocyte tumor development through inhibiting p53 expression and then activating Stat3 phosphorylation [36]. The activation of IRS1 has been shown to affect skin formation and development, being one of the main activators of the differentiation process in skin keratinocytes [37]. E2F1 has both oncogenic and tumor-suppressive properties in skin carcinomas and induces benign skin papillomas or p53 deficiency [38]. Taken together, the selected hub genes could be important markers for skin diseases caused by dermal toxicants.

## 3. Adverse Effects of Cr on Respiratory System

Respiratory exposure to Cr mainly occurs in industrial workers [39]. The inhalation of Cr(VI) is associated with an increased risk in lung cancer, based on numerous observations in occupational epidemiology studies. Among workers with increased lung cancer risk, nasal irritation and perforation have also been reported [40]. Furthermore, lung cancer occurred more often in workers in the chromate-producing industry than in control subjects [41]. Although the mode of action for Cr(VI)-induced lung tumors has not yet been established, these observations would generally support a relationship between inhalation of Cr and lung cancer.

To understand the toxic mechanisms of Cr inhalation, we conducted molecular pathway analysis with Pathway Studio software using the gene expression profile dataset for lung exposure to Cr from the public gene expression database. Microarray datasets GSE24025 and GSE36684 were from the in vitro study of the human bronchial epithelial cell line (BEAS-2B) exposed to Cr(VI) from the GEO database. By the integration and batch effect correction of the different datasets, we investigated the toxic effects of Cr exposure on the respiratory system. Figure 3a shows that exposure to Cr and Cr compounds through inhalation mainly induces DNA damage, cell cycle progression and metastasis, tumor growth, and various respiratory diseases, such as pneumonia, lung neoplasm, pulmonary fibrosis, and lung cancer. Seidler et al. revealed that inhaled Cr(VI) might act by a non-thresholded genotoxic mechanism, form DNA adducts, and induce lung tumor formation [42].

We performed a GO term analysis using all DEGs to predict the biological functions related to Cr inhalation by identifying the significantly enriched BP, CC, and MF terms (Figure 3b). Among the BP terms, ribosomal RNA (rRNA) biogenesis, rRNA metabolic process, and ribonucleoprotein complex biogenesis were significantly enriched. Ye and Shi demonstrated that ribosomal activity was increased in Cr-exposed lung cells, thereby initiating a reconstruction process through the upregulation of protein synthesis [43]. In our study, cell–substrate adhesion and the positive regulation of macrophage cytokine production were suppressed BP terms. The alteration of cell adhesion promotes cancer cell invasion and extravasation in various organs [44,45]. Among the CC terms, pre-ribosome-related terms were enriched, whereas focal adhesion and cell–substrate junction were suppressed. The enrichment of pre-ribosomal processes involved in ribosome biogenesis and protein synthesis supported the enriched BP terms. The suppression of the CC term cellular adhesion is also consistent with the BP result. Among the MF terms, tRNA binding was significantly enriched, while cell adhesion mediator activity was suppressed. The activation of tRNA binding lead to protein synthesis, which is consistent with the BP and CC results. The suppression of cell adhesion mediator activities among the MF terms is also consistent with the previous results. As a result of the GO term analysis, the enrichment of protein synthesis and suppression of cell adhesion were predicted to be the main toxic effects of Cr inhalation.

A network-based gene–gene interaction analysis was performed to select the hub genes associated with Cr inhalation. To analyze the topological parameters among the genes related to Cr inhalation, the gene–gene interaction network was constructed (Figure 3c). We identified the hub genes and core network associated with Cr-induced respiratory effects (Figure 3d). TLR4, TGM2, TNFRSF11B, BDKRB1, KIT, and ZEB1 were the hub genes, and tumorigenic cell processes and diseases were identified as the toxic effects of Cr inhalation. The TLR4 expression in alveolar and bronchial epithelial cells plays a critical role in response to endotoxin [46]. According to Zhang et al., TLR4 deficiency induced lung damage and emphysema [47]. TGM2 is generally overexpressed in lung cancer cells. TGM2-overexpressing cell lines showed high resistance to drugs such as cisplatin in non-small-cell lung cancer [48]. TNFRSF11B, a member of the TNF receptor superfamily, can inhibit tumor apoptosis by binding to a TNF-related apoptosis-inducing ligand. Overexpression of TNFRSF11B can promote tumor growth and metastasis [49]. KIT protein is expressed in small-cell lung cancer, and its kinase activity has been implicated in the pathophysiology of many tumors [50]. KIT expression was associated with advanced disease and a decreased incidence of bone metastasis [51]. According to Larsen et al., increased expression of ZEB1 is associated with tumor grade and metastasis in lung cancer, likely due to its role as a transcription factor in epithelial-to-mesenchymal transition [52]. The selected hub genes could be important markers for respiratory diseases and lung cancer caused by Cr inhalation exposure.

## 4. Adverse Effects of Cr on Gastrointestinal Tract

Cr is a well-known carcinogen in the respiratory tract and can cause lung cancer in occupational workers. In contrast, the human evidence for the toxicity of Cr ingested orally is limited. Beaumont et al. documented increased cancer risks in a Chinese population exposed to Cr(VI) in drinking water [53]. The primary route of exposure in non-occupational workers is food ingestion. The California Environmental Protection Agency (CalEPA) environmental health hazard assessment set a public health goal for total Cr in drinking water. Results from several genotoxicity assays associated ingested Cr(VI) with forestomach tumors in mice.

To understand the toxic mechanisms of Cr ingestion, we conducted a molecular pathway analysis with Pathway Studio software. Microarray dataset GSE120146 was derived from an in vivo study of mouse duodenum exposed to Cr(VI) from the GEO database. Figure 3a shows that ingestion exposure to Cr and Cr compounds mainly induces DNA damage, metastasis, and tumorigenic cell processes. Male infertility, renal disease, and gastrointestinal tract diseases, such as small intestine neoplasm and squamous cell carcinoma, were associated with Cr ingestion. Ingested Cr is able to induce oxidative stress via multiple pathways. Depending on the levels of reactive oxygen species produced, Cr-induced oxidative stress may lead to cell death or tumor formation [53].

We performed a GO term analysis using all DEGs to predict the biological functions related to Cr ingestion by identifying the BP, CC, and MF terms (Figure 4b). Among the BP terms, nucleotide biosynthesis and nucleoside phosphate biosynthesis process were highly enriched. The upregulation of nucleotide biosynthesis can cause cell proliferation and cancer [54]. The suppressed BP terms were alcohol metabolic process, steroid metabolic process, and lipid metabolic process. Those processes were related to liver function, showing that there was an abnormality in liver function due to gastrointestinal tract exposure [55]. Various cell division-related CC terms such as spindle, midbody, kinetochore, and condensed chromosome were enriched. The acceleration of the cell cycle and cell division can lead to tumor proliferation in various digestive cancers [56]. The cell projection membrane, brush border, and brush border membrane were suppressed by Cr ingestion. The brush border can be found in the small intestine and large intestine, where absorption occurs [57]. Impairment of the brush border membrane can cause dysfunction of the digestive tract, and brush border proteins involved in cell polarity have an important role in intestinal tumor development [58]. Among the MF terms, rRNA binding was enriched, while tough symporter and sterol transfer activity were suppressed. Enrichment of ribosomal RNA binding can lead to protein synthesis, consistent with enriched biosynthesis-related BP terms. Additionally, impaired symporter activity and sterol transfer activity are associated with liver dysfunction, one of the typical toxic effects of digestive toxicity. As a result of the GO term analysis, the enrichment of diverse biosynthesis processes and diverse biological indicators of liver dysfunction are predicted to be the main toxic effects of Cr ingestion.

A network-based gene–gene interaction analysis was performed to select the hub genes associated with Cr ingestion. To analyze the topological parameters among the genes related to Cr ingestion, the gene–gene interaction network was constructed (Figure 4c). We identified the hub genes and core network associated with Cr ingestion (Figure 4d). VEGFA, EGFR, APP, SGK1, JUN, and TLR2 were the hub genes, and oxidative stress, cell proliferation, and blood vessel development were cell processes closely linked with Cr ingestion. According to the molecular network, colorectal cancer can occur due to Cr ingestion. VEGFA is a known stimulator of tumor angiogenesis. The positive correlation between VEGFA expression and gastric cancer was verified, and a high level of VEGFA frequently predicted shorter survival time [59]. EGFR is a well-known growth factor receptor involved in the development and progression of carcinogenesis [60]. In the gastrointestinal tract, EGFR overexpression was a prognostic factor in patients with gastric cancer. APP has been associated with cell adhesion, cell motility, and cell proliferation, and APP proteins are highly expressed in gastrointestinal tumors [61]. SGK1 expression favors the development of intestinal tumors in mice [62]. JUN overexpression is observed in various cancers, such as non-small-cell lung cancer, breast cancer, and vulvar cancer, but there is no direct evidence linking JUN overexpression with gastrointestinal cancer. However, JUN overexpression leading to cell cycle progression and metastasis may generate or accelerate cancer [63]. TLR2 has been reported to activate the PI3K/Akt/GSK3β pathway, leading to anti-inflammatory effects in the gut epithelia [64]. The selected hub genes could be important markers for gastrointestinal tract diseases and colorectal cancer caused by Cr ingestion exposure.

## 5. Adverse Human Health Effects of Cr Depend on the Exposure Route

We explored the toxic effects of Cr exposure dependent on the exposure route. We endeavored to organize the toxicity mechanisms, GO terms (BP, CC, and MF), and hub genes associated with Cr exposure (Table 1). DNA damage was found to be a common potential toxic mechanism of Cr for all exposure routes. The relationship between Cr exposure and DNA damage has been well-documented. Cr produces free radicals in the cytoplasm and nucleus, and free radicals can bind to DNA [65]. In human fibroblasts, Cr exposure induced DNA damage by the formation of a sub-G1 peak and arrest of the G2/M phase of the cell cycle [66]. Furthermore, lung exposure to Cr induced DNA damage, tissue irritation, inflammation, cytotoxicity, and, ultimately, lung cancer [67]. Ingestion of Cr through drinking water can form Cr-DNA adducts and induce DNA damage [68]. Metastasis, which was also found to be a common potential toxic mechanism of all exposure routes of Cr, is well-known to be involved in the development of various diseases and cancers. In addition, the results of the GO term analysis identified the disruption of cell adhesion-related BP, CC, and MF terms in all exposure routes. Based on previous studies and our analysis result, we present Cr-DNA adducts and metastasis as a common toxic mechanism of Cr exposure.

An allergic reaction is a potential toxicity mechanism specific to Cr dermal exposure. Although it is known that Cr can penetrate skin tissue and induce an allergic reaction and dermatitis, the underlying mechanism has not yet been clearly verified. Various biosynthesis processes that can cause cell proliferation were enriched in response to Cr dermal exposure. The cell cycle alteration is identified as a specific potential mechanism associated with the potential respiratory effects of Cr inhalation. Moreover, the enrichment of the ribosomal biogenesis process and suppression of cell adhesion are strongly associated with cancer development. Our analysis result found no specific toxicity mechanism related to Cr ingestion. However, enrichment of the nucleotide biosynthesis process and liver function failure was detected as a Cr ingestion-related BP terms.

## 6. Conclusions

Heavy metals have a widespread use in diverse fields, and their environmental impacts and potential health effects have raised serious concerns. Cr is commercially used in various products, and Cr exposure in workers and the general population can occur via various routes [69]. The most prevalent oxidation forms of Cr are Cr(III) and Cr(VI); both forms have been reported as potentially toxic and direct toxicities. In this review, we present the potential toxic mechanism, BP terms, and hub genes for each exposure route by bioinformatic analysis based on public gene expression data. DNA damage was shown as a common potential toxic mechanism of Cr for all three exposure routes, although the detailed BP terms differed. In particular, for the inhalation and ingestion exposure routes, the BP terms identified were associated with cancer, which is intimately linked to the commonly known Cr-induced occupational diseases. We also present hub genes for each Cr exposure route as potential candidates for markers of Cr toxicity. As our analysis was conducted using only partial data, more public genomic data need to be integrated. In summary, diverse bioinformatic analyses to comprehensively understand these complex mechanisms will be beneficial for understanding the Cr-induced toxic effects following various routes of exposure.

## Figures and Tables

**Figure 1 ijms-24-03410-f001:**
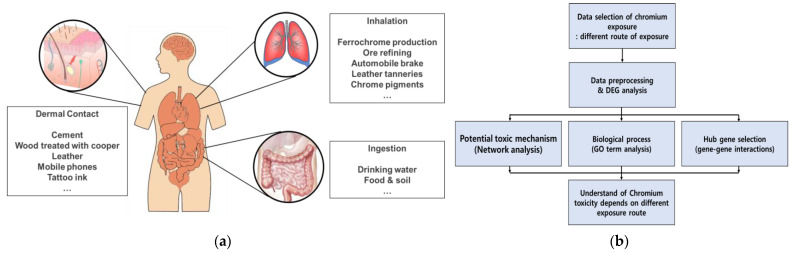
(**a**) Sources of Cr exposure based on different exposure paths. (**b**) Flow chart of the steps used in this review.

**Figure 2 ijms-24-03410-f002:**
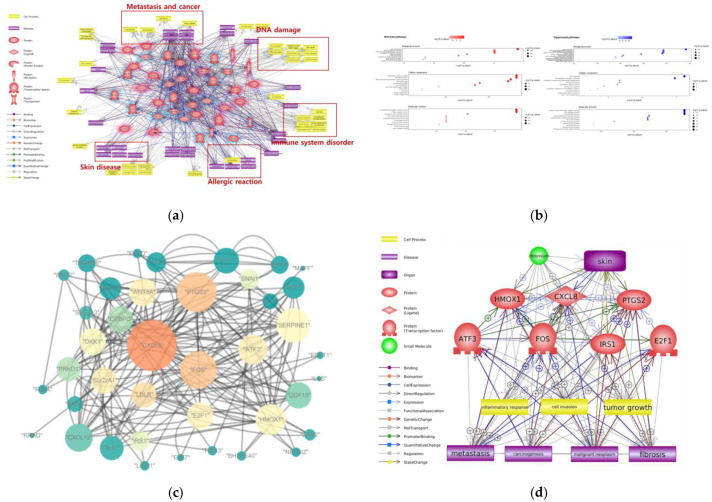
Bioinformatic analysis about Cr toxicity on skin. (**a**) Molecular network includes gene, cell process, diseases associated with Cr dermal contact. Potential toxicity mechanism highlighted with red color box. (**b**) Enriched and Suppressed Biological processes, Cellular components, Molecular functions associated with Cr dermal contact (**c**) Gene–gene interaction network following Cr dermal contact; each node’s color represents the betweenness centrality, each node’s size represents degree of the nodes; (**d**) Summarized molecular network and hub genes about Cr dermal contact.

**Figure 3 ijms-24-03410-f003:**
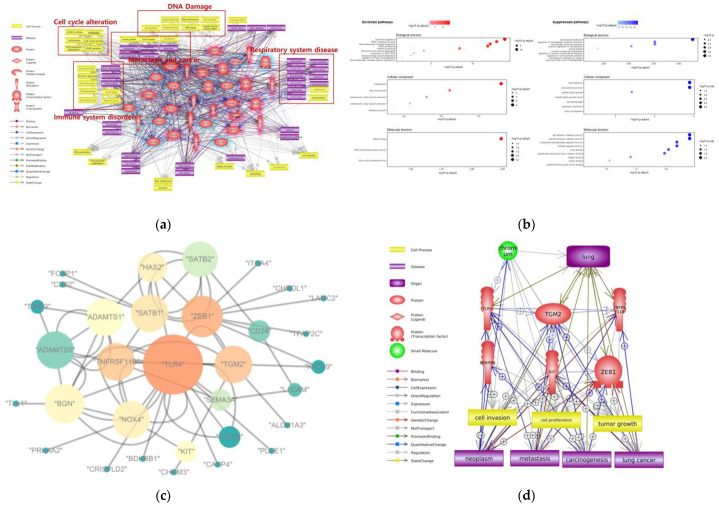
Bioinformatic analysis about Cr toxicity on respiratory system. (**a**) Molecular network include genes, cell processes, diseases associated with Cr inhalation; potential toxicity mechanism highlighted with red colored box; (**b**) Enriched and Suppressed biological processes, cellular components, molecular function about Cr inhalation; (**c**) Gene–gene interaction network following Cr inhalation; each node’s color represents the betweenness centrality, each node’s size represents degree of the nodes; (**d**) Summarized molecular network and hub genes following Cr inhalation.

**Figure 4 ijms-24-03410-f004:**
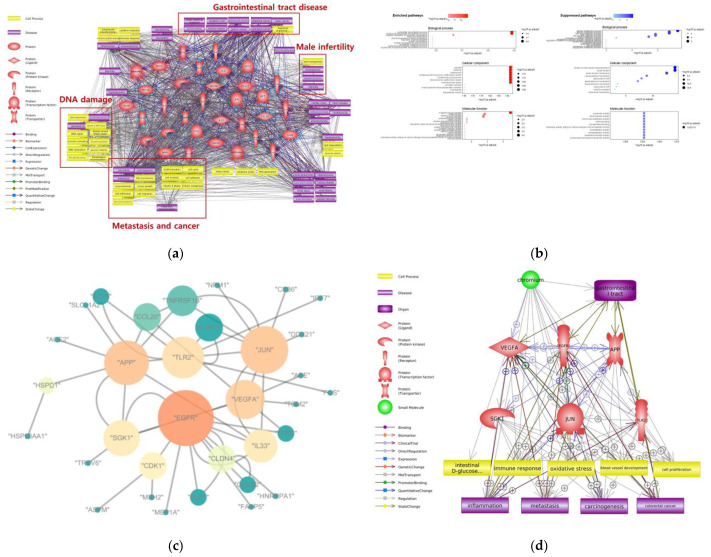
Bioinformatic analysis about Cr toxicity on gastrointestinal tract. (**a**) molecular network includes genes, cell processes, diseases about Cr ingestion; potential toxicity mechanism highlighted with red colored box. (**b**) Enriched and Suppressed biological processes, cellular components, molecular functions associated with Cr ingestion. (**c**) Gene–gene interaction network following Cr ingestion; each node’s color represents betweenness centrality, and each node’s size represents the degree of the nodes; (**d**) Summarized molecular network and hub genes about Cr ingestion.

**Table 1 ijms-24-03410-t001:** Potential toxicity mechanism, biological processes, and hub genes of Cr exposure dependent on different exposure routes.

Exposure Route	Potential Toxicity Mechanism	Biological Process (BP)	Hub Genes
Dermal contact	DNA damage, metastasis, immune system disorder, allergic reaction	RNA modification, sterol biosynthesis process, DNA replication, intrinsic apoptotic signaling pathway	CXCL8, PTGS2, FOS, HMOX1, ATF3, IRS1, E2F1
Inhalation	DNA damage, metastasis, cell cycle alteration, immune system disorder	Ribosome biogenesis, rRNA metabolic process, cell–substrate adhesion, macrophage cytokine production	TLR4, TGM2, TNFRSF11B, BDKRB1, KIT, ZEB1
Ingestion	DNA damage, metastasis	Nucleotide biosynthesis process, nucleotide phosphate biosynthesis process, alcohol metabolic process, steroid metabolic process	VEGFA, EGFR, APP, SGK1, JUN, TLR2

## Data Availability

No new data were created or analyzed in this study. Data sharing is not applicable to this article.

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
