# Peer review of "Adverse Human Health Effects of Chromium by Exposure Route: A Comprehensive Review Based on Toxicogenomic Approach"

_ijms, 2023, doi:10.3390/ijms24043410_

Round 1
Reviewer 1 Report
General:
The manuscript gathers the most recent data available in scientific literature and databases on toxicity mechanisms of different Cr exposure routes, with an interesting genomic analysis of the phenomenon. Nonetheless, in some paragraphs the excessive (and sometime unnecessary) use of acronyms makes reading cumbersome and not very fluent
The title is unclear, probably affected by some typo
the bibliography is updated, with the highest percentage of citations referring to the last 5 years and a few more dated exceptions (the oldest dating back to 1994)
figures 2, 3, and 4 are very interesting and appropriate, but it is mandatory to increase their definition: if enlarged for a better understanding, the detail is lost and the text becomes illegible (sp A and B)
Special:
11-17: please, check the paragraph, sometimes it seems that some words are missing…
45: if we talk about toxicology, the exposure routes are always the natural ones (oral, cutaneous and respiratory), these are not the prerogatives of Cr, and the curiosity arises to know the routes other than the "three main exposure paths"
113 and forward: please, specify the acronym at its first use
112-132, 188-204, 254-274: see general comment on excessive acronyms use
233-234: the whole previous section deals about this, the remark is unnecessary
Author Response
General:
The manuscript gathers the most recent data available in scientific literature and databases on toxicity mechanisms of different Cr exposure routes, with an interesting genomic analysis of the phenomenon. Nonetheless, in some paragraphs the excessive (and sometime unnecessary) use of acronyms makes reading cumbersome and not very fluent.
The title is unclear, probably affected by some typo. the bibliography is updated, with the highest percentage of citations referring to the last 5 years and a few more dated exceptions (the oldest dating back to 1994). figures 2, 3, and 4 are very interesting and appropriate, but it is mandatory to increase their definition: if enlarged for a better understanding, the detail is lost and the text becomes illegible (sp A and B).
Response: We greatly appreciate your valuable feedback. We corrected the typos in the title. We changed the title to “Adverse Human Health Effects of Chromium by Exposure Route: A Comprehensive Review Based on Toxicogenomic Approach”. The references have been updated to the latest ones, and the resolution of the picture images has been increased. Also, English revision was conducted for the entire manuscript.
11-17: please, check the paragraph, sometimes it seems that some words are missing…
Response: We checked the paragraph and added an approach to the underlying mechanism of Cr exposure.
45: if we talk about toxicology, the exposure routes are always the natural ones (oral, cutaneous and respiratory), these are not the prerogatives of Cr, and the curiosity arises to know the routes other than the "three main exposure paths".
Response: Thank you for your insightful comment. We modified content that could cause potential confusion.
113 and forward: please, specify the acronym at its first use.
Response: We added the description of gene ontology.
112-132, 188-204, 254-274: see general comment on excessive acronyms use.
Response: As you commented, we reduced the use of acronyms.
233-234: the whole previous section deals about this, the remark is unnecessary.
Response: As you commented, we deleted the redundant explanation about Cr inhalation exposure.
Reviewer 2 Report
Chromium(VI) is highly toxic pollutant. The impact of chromium on public health is an important issue.
I have reviewed the manuscript entitled: .
„ A review of adverse effects of chromium depends on different exposure routes: based on toxicogenomic approach”.
In my opinion the manuscript need minor revision.
Comment1
Line 35 “Cr is abundant in the Earth’s crust” What is the chromium content of the earth's crust?
Comment 2
Line 46, “A large proportion of the population” please specify this value in percentage
Comment 3
Line 83 figure 1 is not captioned
Comment 4
Line 85 What are the concentrations of chromium(VI) or chromium(III) compounds in cosmetics?
Comment 5
Line 161 figure 2 A and B is unreadable
Comment 6
Line 113 Please explain the abbreviation for GO
Comment 7
Line 225 figure 3 A and B is unreadable
Comment 8
Line 232 and 168 Two sections have number 3? needs to be renumbered
Comment 9
Line 244 you need to remove the period
Comment 10
Line 297 figure 3 B is unreadable
Author Response
Chromium (VI) is highly toxic pollutant. The impact of chromium on public health is an important issue. have reviewed the manuscript entitled:
„ A review of adverse effects of chromium depends on different exposure routes: based on toxicogenomic approach”. In my opinion the manuscript need minor revision.
Response: We greatly appreciate your valuable feedback. We revised our manuscript following your comment. These changes are marked with“Track Changes” function in the revised manuscript.
Comment1: Line 35 “Cr is abundant in the Earth’s crust” What is the chromium content of the earth's crust?
Response: The Cr concentration of the Earth’s crust is 100 μg/g, mainly composed of Cr(III) (Nriagu et al. 1988). We included this information in our revised manuscript.
Comment 2: Line 46, “A large proportion of the population” please specify this value in percentage.
Response: This content was written to explain the non-occupational Cr exposure of the general population. We modified content that could cause potential confusion.
Comment 3: Line 83 figure 1 is not captioned.
Response: The caption of Figure 1 has been added.
Comment 4: Line 85 What are the concentrations of chromium (VI) or chromium (III) compounds in cosmetics?
Response: According to Bocca et al. (2014), toxicity caused by Cr(VI) is stronger than that caused by Cr(III) in cosmetics. Cr was detected at concentrations up to 318 μg/g in cosmetics, such as eye shadows, and both Cr(III) and Cr(VI) were detected. We included information about Cr-containing products in our revised manuscript.
Comment 5: Line 161 figure 2 A and B is unreadable.
Response: We replaced Figure 2 with a higher-resolution image.
Comment 6: Line 113 Please explain the abbreviation for GO.
Response: We added the description of GO.
Comment 7: Line 225 figure 3 A and B is unreadable.
Response: We replaced Figure 3 with a higher-resolution image.
Comment 8: Line 232 and 168 Two sections have number 3? needs to be renumbered.
Response: We renumbered the relevant sections.
Comment 9: Line 244 you need to remove the period.
Response: As you commented, we modified content that could cause potential confusion.
Comment 10: Line 297 figure 3 B is unreadable.
Response: We replaced Figure 3 with a higher-resolution image.